# Electrochemical Evaluation of Directly Electrospun Carbide-Derived Carbon-Based Electrodes in Different Nonaqueous Electrolytes for Energy Storage Applications

**Siret Malmberg [1,2,\*], Mati Arulepp [2], Elvira Tarasova [1], Viktoria Vassiljeva [1], Illia Krasnou [1] and Andres Krumme [1]**

[1] Department of Materials and Environmental Technology, Tallinn University of Technology, Ehitajate tee 5, 19086 Tallinn, Estonia; elvira.tarsova@ttu.ee (E.T.); viktoria.vassiljeva@ttu.ee (V.V.); illia.krasnou@ttu.ee (I.K.); andres.krumme@ttu.ee (A.K.)

[2] Skeleton Technologies OÜ, Valukoja 8, 11415 Tallinn, Estonia; mati.arulepp@skeletontech.com

\* Correspondence: siret.malmberg@ttu.ee

**Abstract:** This study focuses on the electrochemical behavior of thin-layer fibrous carbide-derived carbon (CDC) electrospun electrodes in commercial and research and development stage organic-solvent and ionic liquid (IL) based electrolytes. The majority of earlier published works stated various electrolytes with asymmetric cells of powder-based pressure-rolled (PTFE), or slurry-cast electrodes, were significantly different from the presented CDC-based fibrous spun electrodes. The benefits of the fibrous structure are relatively low thickness (20 μm), flexibility and mechanical durability. Thin-layered durable electrode materials are gaining more interest and importance in mechanically more demanding applications such as the space industry and in wearable devices, and need to achieve a targeted balance between mechanical, electrical and electrochemical properties. The existing commercial electrode technologies lack compatibility in such applications due to their limited mechanical properties and high cost. The test results showed that the widest potential window $dU \leq 3.5$ V was achieved in 1.5 M 1-ethyl-3-methylimidazoliumbis(trifluoromethyl-sulfonyl)imide (EMIm-TFSI) solution in acetonitrile (ACN). Gravimetric capacitance reached 105.6 F g$^{-1}$ for the positively charged electrode. Cycle-life results revealed stable material capacitance and resistance over 3000 cycles.

**Keywords:** electrospinning; electrolyte; carbide-derived carbon; supercapacitor; double-layer capacitance; cycle-life

## 1. Introduction

Supercapacitors, also known as electrochemical double-layer capacitors (EDLC) are energy storage devices with the advantage over conventional capacitors of having significantly higher energy density in a wide range of power capabilities, combined with a long cycle-life [1]. In conventional supercapacitors there are no redox reactions and charge separation occurs upon polarization at the electrode/electrolyte interface, which is also called the electrochemical double layer [2]. The energy storage process in EDLCs is fast and reversible due to the physical adsorption process of electrolyte ions on the carbon surface. However, this is not the case with lithium ion batteries (LIBs), where energy is stored by chemical reactions [3]. Therefore, EDLCs are suitable for a large number of applications where large amounts of energy need to be stored or released in a short time-frame. For example, supercapacitors are primarily used in hybrid electric and fuel cell vehicles like buses [4], trains [5]

and trolley buses [6]. Batteries and fuel cells exhibit energy densities a few orders of magnitudes larger than supercapacitors. Therefore, it is not sufficient to use only supercapacitors as energy sources. Supercapacitors are more beneficial in situations where a great deal of power is needed in a short time, which batteries cannot provide so efficiently [3–6]. Therefore, the use of supercapacitors in combination with batteries provides significant advantages in terms of energy efficiency. For example, almost all the kinetic energy can be stored in the supercapacitor portion of the energy storage system during braking and can be further reused during accelerating. These high-power peaks would otherwise damage LIBs, while fuel cells do not offer any possibility for fast energy recuperation [1]. Other application areas of supercapacitors are energy harvesting systems [7], solar arrays [8] and windmill turbines [9], in which the supercapacitors' high energy efficiency and lifetime has proven to be advantageous [1].

The overall performance of supercapacitors is influenced by the interactions between the electrode and the electrolyte inside the cell. Besides the operative voltage, electrolytes have great influence on the other parameters of EDLCs such as equivalent series resistance and power density, cycling stability, operating temperature range and self-discharge rate. The electrostatic interactions between electrolyte ions and the surface of carbon electrodes thus play an important role in impacting the overall EDLC characteristics. Generally, a larger accessible surface area leads to higher energy density [10].

In general, an ideal electrolyte for EDLC applications should exhibit some critical characteristics: high electrochemical stability, i.e., broad electrochemical potential window; a wide working temperature range; high ionic conductivity; high polarity; low viscosity; environmental friendliness; low flammability and low cost [11]. Electrolytes are categorized into several groups such as aqueous; organic; IL; solid-state and redox-active electrolytes [11,12]. The most used type of electrolyte in industrial supercapacitors is a solution of quaternary ammonium salt in organic solvent, often acetonitrile (ACN) or propylene carbonate (PC) [1,6]. Acetonitrile-based electrolytes show higher conductivity and lower viscosity compared to PC-based electrolytes [13]. Therefore, EDLCs containing ACN-based electrolytes out-perform those containing propylene carbonate, especially regarding power density and specific energy at lower temperatures. However, PC-based electrolytes are considered safer alternatives compared to the acetonitrile-based electrolytes due to a lower level of toxicity and higher flash point [11]. The maximum operative voltage of the majority of nonaqueous electrolyte based commercial supercapacitors is between 2.5–3.0 V.

IL-based electrolytes are widely used for increasing the upper voltage limit of EDLCs [2,14]. Disadvantages of ILs are their sensitivity to moisture, moderate conductivity as a result of high molecular weight and viscosity [5], and high cost, impacting their cost per function [8,9]. In addition, moisture can easily affect the stability and conductivity of an IL. Even a small amount of water can dramatically reduce the electrochemical stability of ILs.

Typically, industrial supercapacitors utilize porous carbon electrodes with high specific surface areas. Over the last decade, several carbon candidates such as fullerenes [15], carbon nanotubes [16]; graphene [17] and CDCs [18], have been reported for electric double layer capacitor applications. Each of these compounds presents exceptional properties for capacitor electrodes. Their theoretical capacitance in a two electrode configuration ranges from 60 up to 360 F g$^{-1}$ [19–22]. Carbide-derived carbons differ from other available carbon materials due to their unique nanoporous structure, narrow pore size distribution and the possibility of fine tuning of the pore size [10,21–24]. CDCs have been under increased attention over the past decade as high capacitance materials due to unique properties. Polytetrafluoroethylene or polyvinylidene fluoride binders based CDC electrodes have been tested with a variety of organic electrolytes including IL-based electrolytes [10,25–27]. The highest reported specific capacitance of CDC was up to 190 F g$^{-1}$ with a 1-ethyl-3-methylimidazolium tetrafluoroborate (EMIm-BF$_4$)-based electrolyte [10]. As mentioned before, EDLCs store energy in the electrochemical double layer formed between the carbon electrode's surface and the electrolyte's ions. Generally, higher specific surface areas in carbon materials show higher gravimetric capacitances. However, carbon materials have natural limits to their possible capacitance, as their active surface area per weight is limited. Still, one can utilize the available surface most efficiently by matching carbon pore size with

the ion size of an electrolyte, ensuring access to the carbon surface by having enough transport channels and, with decreasing distance between charges, in the double-layer [12,22]. Otherwise, the electrical resistance of the electrode increases due to limited access of the electrolyte to the pores [28].

With porous carbon, the capacitance can depend on varying the different electrolytes with different ion sizes. To achieve high energy, it is important to know both the dominant pore size of the carbon and the size of the electrolyte ions. The $N_2$ adsorption method, or density functional method, are used for investigating pore sizes of carbon [21].

The ion sizes of electrolytes can be calculated by various methods, such as determining the bare molecular and solvated ion sizes [12]. It is important to note that the dimensions of the solvated ions depend on the concentration of the solution in the case of electrolyte solutions. This is particularly important in the case of ILs, where the electrolyte ions may also be unsolvated, such as in the case of pure IL. Lin et al. found that although the bare ion sizes of $EMIm^+$ and $TFSI^-$ were similar, they were nevertheless different for solvated ions in the acetonitrile solvent: (TFSI/ACN < $EMIm^+$/ACN) [29]. Quaternary ammonium salts' cation size decrease in the following order: tetraethylammonium, spiro-(1,1')-bipyrrolidinium and triethylmethyl-ammonium ($TEA^+$ > $SBP^+$ > $TEMA^+$), calculated for the solvated ion radii [2,30].

Carbon electrodes offer good electrical conductivity and a high surface area. Carbon-saturated electrospun fibrous electrodes can be used as alternatives to rolled or casted porous carbon electrodes. Recently, electrospun materials have been shown to also provide high EDL-capacitance with good flexibility and good mechanical properties [31].

Electrospinning is the process of forming fibers of submicrometer diameter by an electrical field. It is a scalable technique whereby a drop of polymer from the tip of a needle is drawn into fibers by electrostatic force. The solvent is evaporated and fibers are formed on their way to the current collector [32].

Numerous attempts have been made to increase the voltage stability range of carbonaceous electrodes in aqueous and nonaqueous electrolytes [18,19]. However, little is reported towards increasing the working potential of electrospun fibrous electrodes in supercapacitors [33,34]. Levitt et al. showed capacitance up to 205 mF $cm^{-2}$ with composite MXene/PAN fibers, which is three times higher than pure carbonized PAN [33]. Another approach for making fibrous electrodes has been done with a PANi/carbonized polyimide combination [34]. However, the carbonization process itself is destructive and certainly it has a relatively high cost from a large-scale production point of view.

The present study analyzed the electrochemical behavior of electrospun fibrous electrodes saturated with CDC in different organic and IL-based electrolytes. A three-electrode test setup was used to explore the maximum working potential of the electrodes in different electrolytes as well as the respective electrodes' capacitances. The values presented herein can thus serve to design a two-electrode system utilizing the respective electrospun electrodes and investigated electrolytes. Moreover, the study also outlines the importance of electrolyte selection for the optimization of any EDLC's electrochemical performance.

## 2. Experimental Section

### 2.1. Materials and Processes

For analysis of the electrochemical behavior of the fibrous electrodes in various electrolytes, the best performing recipe from a previous study was used [31]. Titanium carbide (TiC) based CDC material was used as the main capacitive component. For this approach, a TiC precursor was converted to CDC by using $Cl_2$ treatment at 900 °C, followed by a hydrogen gas purification step at 800 °C to remove residues of chlorine. CDC was produced by the company Skeleton Technologies OÜ. The specific surface area of synthesized CDC was 1577 $m^2$ $g^{-1}$ with an initial particle size around 1–5 μm. The CDC particles were milled according to the procedure described in our previous study [31].

A mixture of milled CDC particles and carbon black (CB, Timcal, Deutschland GmbH, Düsseldorf, Germany) with weight (wt) % ratio of 80/20, respectively, was dispersed by 2 h ultrasonic treatment in dimethylformamide (DMF, Sigma Aldrich, Tartu, Estonia). The resulting mixture of the solvent and carbon particles was mechanically stirred at 40 °C for 24 h. Thereafter, 7 wt% of polyacrylonitrile (PAN, Sigma Aldrich, Mw = 150,000 g mol$^{-1}$) was added to the solution, while the ratio of PAN and carbon was 50/50. The whole solution was mechanically stirred for another 24 h at 40 °C. As a last step before the electrospinning process, 15 wt% of IL EMIm-BF$_4$ was added to the solution and mixed for 0.5 h at 40 °C [31]. The purpose of IL was to increase the solution's conductivity for a successful electrospinning process.

The prepared solution was electrospun by using a horizontal electrospinning system. The electrospinning conditions were as follows: the pumping rate of the solution was 0.5 mL h$^{-1}$, the distance between the spinneret and drum collector was 8 cm and a DC voltage of 15 kV was applied [31]. Scanning electron microscopy (SEM) of the prepared electrode, with two different magnifications, was carried out with a Gemini Zeiss Ultra 55 and is presented in Figure 1. SEM analysis displayed the fibrous structure of the electrode. The distribution of CDC/CB particles was clearly observed on the surface and inside of the main PAN/DMF fibers. As seen on the SEM images the contact between the CDC/CB particles remained, which is important to achieve good conductivity of the fibers. More detailed SEM analysis of the samples is described in our previous work [31].

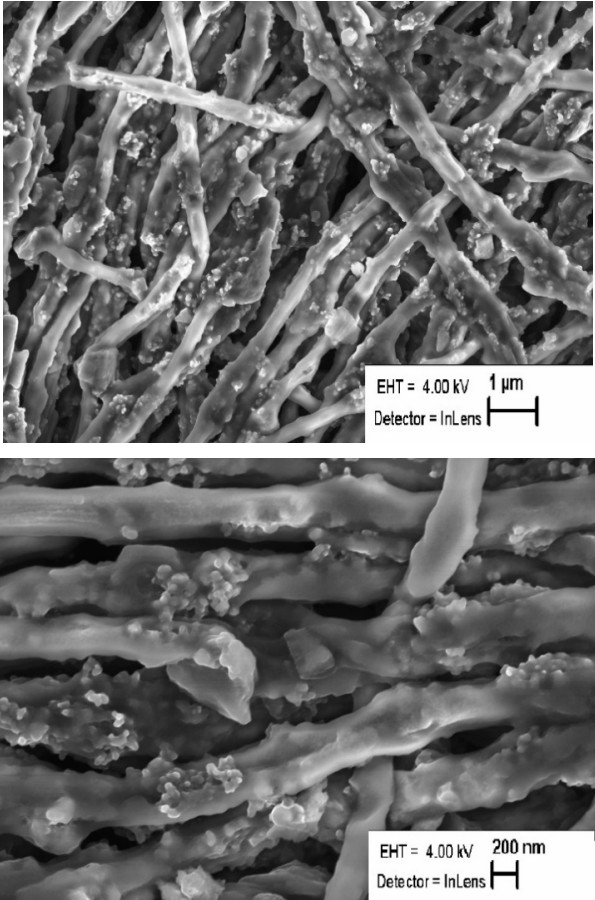

**Figure 1.** Different magnifications of scanning electron microscope (SEM) images of electrospun carbide-derived carbon (CDC) based fibrous electrode.

The thermal stability of the electrode was evaluated by thermal gravimetric analysis (TGA) by Labsys Evo TG DTA Omni Star. As shown in Figure 2, the electrospun fibrous electrode sample decomposed in three steps. The first mass loss stage is present between 247–317 °C, which has an

early shoulder at 290 °C, which indicates several parallel or competing reaction kinetics. The main reaction at this stage is the degradation of PAN, which occurs at 260 °C. This is comparable with PAN degradation found in V. Salles, etc. research, where pure polyacrylonitrile fibers decomposed between 250–300 °C [35]. The second mass loss stage starts at 318 °C and ends at 442 °C, indicating the degradation of ionic liquid EMImBF$_4$, which, based on the literature, has been found to occur between 333 and 455 °C [36].

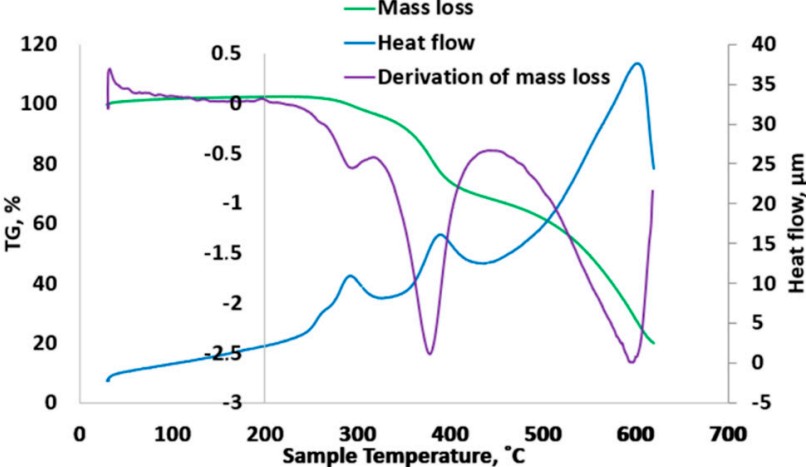

**Figure 2.** Decomposition characteristics of fibrous electrospun electrode material.

The degradation of ionic liquid also shows the highest heat flow. Third mass loss stage with the highest mass loss of 48.3%, which has a peak at 600 °C is the degradation of carbonous fillers CDC and CB. The porosity characteristics of the CDC powder was determined from N$_2$ adsorption at −196 °C using the NOVA touch LX2 (Quantachrome Instruments, Boynton Beach, FL, USA). Before measuring, the samples were dried for 12 h in vacuum at 300 °C. The Brunauer-Emmmet-Teller( BET) surface area was calculated from N$_2$ adsorption according to BET theory at a pressure interval P/P$_0$ of 0.02–0.2. The micropore volume (V$_\mu$), calculated by a t-plot model and the total pore volume (V$_{tot}$) was calculated at P/P$_0$ of 0.97. Calculations of pore size distribution (PSD) were done by using a quenched solid density functional theory (QSDFT) equilibria model for slit type pores. Before measurement, the sample was degassed in a vacuum at 300 °C for 12 h. The porosity characteristics and PSD are presented in Table 1 and in Figure 3.

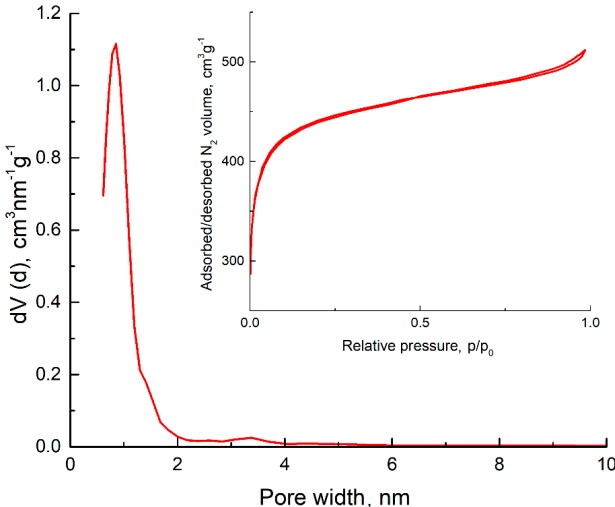

**Figure 3.** The pore size density (PSD) and N$_2$ absorption/desorption isotherm of tested CDC samples.

**Table 1.** Porosity characteristics of CDC particles.

| Sample | $S_{BET}$, m$^2$ g$^{-1}$ | $V_\mu$, cm$^3$ g$^{-1}$ | $V_{tot}$, cm$^3$ g$^{-1}$ | APS nm |
|--------|--------|--------|--------|--------|
| CDC | 1560 | 0.71 | 0.82 | 0.95 |

For electrochemical tests, the prepared electrodes were cut with a diameter of 6 mm and an average coat weight of 1.86 g m$^{-2}$ and dried under vacuum at 100 °C for 24 h to remove absorbed gases and water. For testing in various electrolytes, the weights of the electrodes were recorded after drying with an analytical scale (Mettler AE 163, Tallinn, Estonia). Afterwards, the prepared electrodes were assembled into three-electrode test cells with a glass fiber separator (purchased from VWR, Dresden, Germany), including a large carbon counter and carbon reference electrodes (RE).

### 2.2. Electrolytes

Commercial electrolytes 1.0 M TEA-BF4/ACN and 1.8 M TEMA-BF4/ACN were purchased from Honeywell. 1.5 M SBP-BF4/ACN was purchased from Japan Carlit Co., Ltd, Tokyo, Japan.

ILs, EMIm-TFSI and EMIm-BF$_4$ were purchased from Sigma Aldrich. Both conductive salts were mixed with acetonitrile in a 1.5 molar concentration in a dry box in a nitrogen atmosphere (O$_2$ < 0.5 ppm and H$_2$O < 0.5 ppm) to prevent contamination and to keep electrolytes dry. The physical properties and chemical structure of the electrolytes are presented in Table 2 and Figure 4.

**Table 2.** Physical properties of electrolytes and ions.

| Acronym | Chemical Formula | Ionic Radii, nm | Concentration in Acetonitrile, M (in This Study) | Reference |
|---------|------------------|-----------------|--------------------------------------------------|-----------|
| TEA$^+$ | $(C_2H_5)_4N^+$ | 0.670 | 1 | [37] |
| TEMA$^+$ | $(C_2H_5)_3CH_3N^+$ | 0.327 | 1.8 | [38] |
| EMIm$^+$ | $C_6H_{11}N_2^+$ | 0.326 | 1.5 | [39] |
| SBP$^+$ | $C_8H_{16}N^+$ | 0.420 | 1.5 | [21] |
| BF$_4^-$ | BF$_4^-$ | 0.480 | variable | [29] |
| TFSI$^-$ | $(CF_3SO_2)_2N^-$ | 0.227 | 1.5 | [39] |

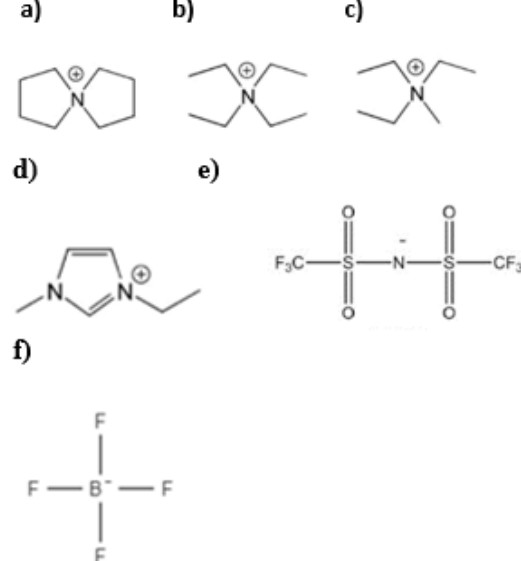

**Figure 4.** Chemical structures of cations and anions: (**a**) SBP, (**b**) TEA, (**c**) TEMA, (**d**) EMIm, (**e**) TFSI, (**f**) BF$_4$ [40,41].

*2.3. Electrochemical Characterization*

The electrochemical characteristics of the electrospun materials were measured using cyclic voltammetry (CV), constant-current charge-discharge (CC), and electrochemical impedance spectroscopy (EIS) methods, all carried out using a Gamry Interface 1010 E equipment, Warminster, Pennsylvania, United States of America.

The CV plots were measured at potential scan rates ($v$) of 50 mV s$^{-1}$ to 1 mV s$^{-1}$. The EDL capacitance was calculated from CV data by dividing measured current values $i$ by the voltage scan rate $v$, according to Formula (1):

$$C = \frac{i}{v} \tag{1}$$

In addition to the information about capacitance, CV represents information regarding electrochemical processes concerning kinetics, cycle efficiency and visible faradaic reactions [42,43]. The coulomb efficiency ($E_q$) is used to characterize charge-discharge cycle efficiency in a CV plot. An efficiency of more than 90% indicates a reversible charge-discharge cycle, where almost all applied charge is returned during discharge. Coulomb efficiency was calculated according to the following formula:

$$E_q = \frac{Q_{chrage}}{Q_{discharge}}, \tag{2}$$

where $Q_{chrage}$ is the amount of electrical charge used during material recharging and $Q_{discharge}$ is the amount of charge received during discharging.

Current charge-discharge is the most common method for measuring integral capacitance. It also correlates most closely with the end application results [44]. The applied potential is selected based on the electrolyte's stability obtained from CV plots. The potential limits were selected for negatively and positively charged systems, −1.2 V and 1.2 V, respectively. For the positively charged electrode ($C_A$), and the negatively charged electrode ($C_C$), capacitance values from galvanostatic cycles were calculated at a discharge current density of 0.5 mA cm$^{-2}$. Since the discharge curves were not linear, the capacitance was calculated by integrating the charges in accordance with Equation (3).

$$C_A; C_C = \int_0^{E_{max}} \frac{idt}{\Delta E} \tag{3}$$

where $\Delta E$ is the potential range for oppositely charged electrodes, 1.2–0 V (positive) and −1.2–0 V (negative) vs. RE [43].

Electrochemical impedance spectra were measured in the AC frequency ranges 200 kHz to 5 mHz, at a fixed electrode potential of −0.5 V vs. carbon reference. To describe the impedance spectroscopy, complex $Z''$, $Z'$ plots known as Nyquist plots are generally presented. Nyquist plots consist of three regions in the case of a porous electrode: the small depressed semicircle at higher AC frequencies, the porous region with a −45° or lower slope and the double-layer capacitance region with a slope of about −90°. In the case of series RC circuits, the total impedance ($Z$) can be described by Equation (4):

$$Z = Z' + Z'' = R + \frac{1}{j\omega C_s}, \tag{4}$$

where $Z'$ is real impedance and $Z''$ imaginary impedance, $j$ is the imaginary number $\sqrt{-1}$, $C_s$ is series capacitance, $\omega$ is angular frequency, $\omega = 2\pi f$, and $f$ is AC frequency in Hz.

The series capacitance values for each electrode material in different electrolytes were calculated by Equation (5).

$$C_s = -\frac{1/Z''}{\omega} \tag{5}$$

$R_s$ can be determined by frequency response analysis based on Equation (6):

$$R_s = Z'$$ 

(6)

The $R_s$ value, at $Z'' = 0$ is the impedance of the electrochemical system.

The phase angle response to frequency for an ideal capacitor is $-90°$. However, it is slightly below $-90°$ in practice, which is frequency dependent, and, therefore, read at the lowest frequency [24,43].

A common method for evaluating long-term stability of a supercapacitor is cycling between maximum and minimum working voltages by the CV or constant current methods for longer periods of time [45]. Therefore, in this study, the cycle-life was carried out at a voltage range of 0 to 2.3 V at a voltage scan rate 20 mV s$^{-1}$. The capacitance retention was evaluated by CV, and resistance increase by EIS.

## 3. Results and Discussion

### 3.1. Galvanostatic Charge and Discharge

The galvanostatic charge-discharge method was performed in the four different electrolytes: two IL and two quaternary ammonium salt-based solutions. To reach the capacitance limits of the positively and negatively charged electrodes, the materials were held at a fixed potential for 5 min before being discharged. Thereafter, the material was recharged for the next cycle. The same profile was repeated five times, firstly at negative and, thereafter, at positive potential regions separately. The average DL capacitance values for the last three cycles are given in Table 3 and discharge curves are shown in Figure 5.

**Table 3.** Capacitance values obtained from constant current discharge plots at $I = 0.5$ mA cm$^{-2}$.

| Electrolyte | Electrode Coat Weight | Positively Charged Electrode | | | Negatively Charged Electrode | | |
|---|---|---|---|---|---|---|---|
| | g m$^{-2}$ | mF cm$^{-2}$ | F cm$^{-3}$ | F g$^{-1}$ | mF cm$^{-2}$ | F cm$^{-3}$ | F g$^{-1}$ |
| EMIm-TFSI/ACN | 2.29 | 20.6 | 10.3 | 89.8 | 20.6 | 10.3 | 89.8 |
| SBP-BF$_4$/ACN | 1.83 | 14.4 | 14.4 | 78.5 | 19.7 | 9.9 | 95.3 |
| TEMA-BF$_4$/ACN | 1.02 | 5.7 | 2.8 | 70.8 | 6.9 | 2.3 | 55.8 |
| EMIm-BF$_4$/ACN | 2.29 | 24.2 | 12.1 | 105.6 | 19.7 | 9.9 | 76.2 |

EMIm-TFSI/ACN: 1-ethyl-3-methylimidazoliumbis(trifluoromethyl-sulfonyl)imide in acetonitrile; SBP-BF4/ACN: spiro-(1,1′)-bipyrrolidinium tetrafluoroborate in acetonitrile; TEMA-BF4/ACN: tetraethylammonium tetrafluoroborate in acetonitrile and EMIm-BF4/ACN: 1-ethyl-3-methylimidazolium tetrafluoroborate in acetonitrile.

The highest capacitance of 105.6 F g$^{-1}$ for a positively charged electrode was achieved with an EMIm-BF$_4$/ACN electrolyte, whereas the capacitance for a negatively charged electrode was only 76.2 F g$^{-1}$ and the difference between the capacitance for a positively and negatively charged electrode was 28%. The lowest capacitance at both potentials was measured in the TEMA-BF$_4$ electrolyte; the capacitance achieved for the negatively charged electrode was 55.8 F g$^{-1}$ and for the positively charged electrode 70.8 F g$^{-1}$. Therefore, it seems that with similar BF$_4^-$ anions, the counter ion still has influence on the material capacitance. ILs are also known to have lower conductivity, but somewhat higher capacitance, as measured for porous carbon electrodes [32]. A similar result was observed here. For SBP-BF$_4$/ACN the capacitance on the positive potential was 17% lower compared to the capacitance on the negative potential. However, the capacitance for the EMIm-TFSI/ACN electrolyte was equal for positive and negative potentials. The capacitance values on negative potentials in SBP-BF$_4$/ACN and for EMIm-TFSI/ACN electrolytes were rather similar, 95.3 F g$^{-1}$ and 89.8 F g$^{-1}$, respectively, most probably due to the similar size of cations.

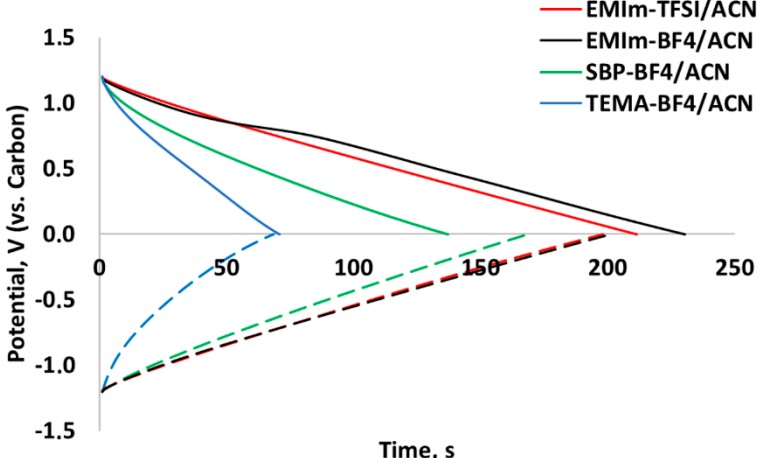

**Figure 5.** Discharge voltage profiles for positively and negatively charged electrodes at a current density of 0.5 mA cm$^2$.

### 3.2. Cyclic Voltammetry

Cyclic voltammetry was applied to investigate the electrochemical stability of fibrous electrodes in different electrolytes. CV plots were measured with voltage scan rates from 5 to 50 mV s$^{-1}$ to evaluate the material stability over time and, more importantly, if the supercapacitor built from CDC-based fibrous electrodes was capable of fast charging rates. Figure 6 presents typical dependence of voltage scan rates for fibrous CDC-based electrodes. As shown on the graph, if the voltage scan rate was increased four times, there was no change in capacitance. In the case of the 10-times increase, capacitance was decreased around 20%.

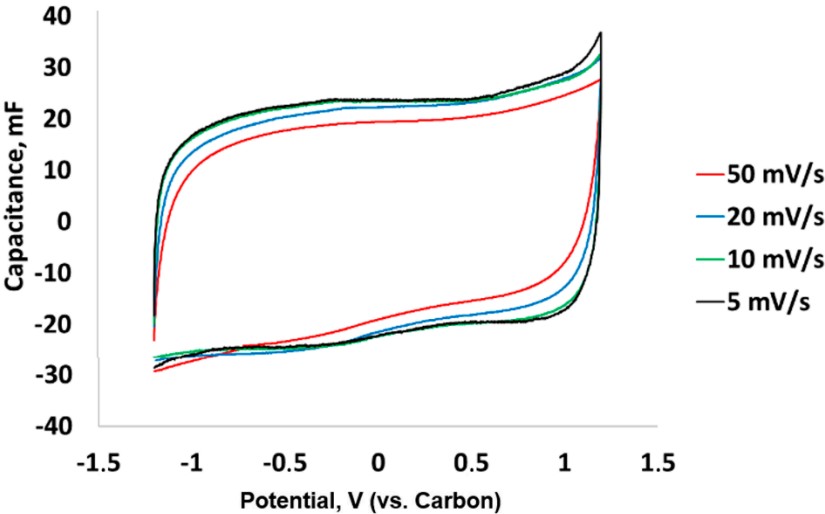

**Figure 6.** Capacitance dependence of voltage scan rates for fibrous CDC-based electrode material in 1.5 M EMIm-TFSI/ACN electrolyte.

The CV method also enabled us to investigate the positive and negative potential limits and coulombic efficiency at different electrode potential intervals of the system. The positive and negative potential limits were determined by the onset of the faradaic current from the nonfaradaic current. The method itself is well described by the studies of Xu [46], Ruschhaupt [45] and Weingarth [47]. The coulombic efficiency and potential windows for positively and negatively charged electrodes are presented in Figure 7A–D.

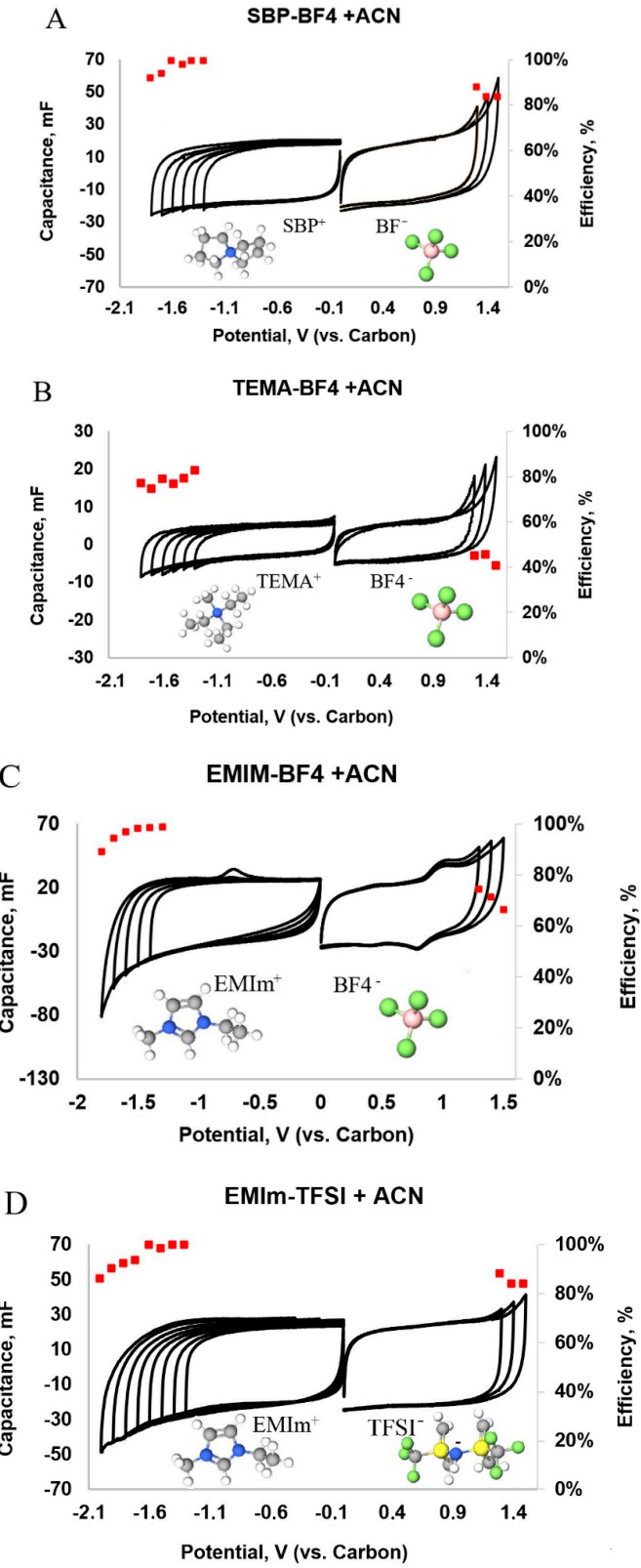

**Figure 7.** Cycling voltammograms expressed as capacitance vs. potential (black lines) and energy efficiency at respective potential (red dots) in electrolytes: (**A**) SBP-BF$_4$/ACN (**B**) TEMA-BF$_4$/ACN, (**C**) EMIm-BF$_4$/ACN and (**D**) EMIm-TFSI/ACN.



Generally, wider potential limits together with higher coulombic efficiency, were observed for negatively charged electrodes compared to the positive potentials (Figure 7). The widest potential window was achieved for the EMIm-TFSI/ACN electrolyte, where negative potential limit reached up to −2.0 V and the positive potential limit reached +1.5 V vs. RE, (total stability voltage dU ~ 3.5 V). The $E_q$ values at negative electrode potentials did not drop below 90%, whereas at positive electrode potentials $E_q$ was almost <85%. With the EMIm-BF$_4$/ACN electrolyte, the decrease of $E_q$ was observed on negative potentials already at −1.6 V and on positive potentials at +0.8 V vs. RE, which is not close to potential limits for this ionic IL-based electrolyte. Furthermore, additional peaks were observed on the voltammogram of negative potentials, where the peak appeared at −0.5 V with the EMIm-BF$_4$/ACN electrolyte. Evolution of the negative discharge peak was observed when negative potential exceeded −1.7 V, which is most probably caused by surface compounds formed with negative overpotential. The peak was reversible on positive potentials close to 0.9 V vs. RE. The origin of this peak is not clear.

A smaller potential window was observed with the TEMA-BF$_4$/ACN electrolyte compared to the other tested electrolytes. The negative limit was −1.9 V, close to the EMIm-TFSI/ACN electrolyte. However, the stable positive potential was reached close to the +1.0 V vs. carbon reference. The efficiency for a positively charged electrode was lower than expected in the TEMA-BF$_4$/ACN electrolyte, probably because CV plots were recorded on a wider electrode potential limits than the EDLC region and also includes parasitic processes. The widest potential window of 3.0 V was observed for the SBP-BF$_4$/ACN electrolyte from the quaternary ammonium salt electrolytes.

### 3.3. Electrochemical Impedance Spectroscopy

The three-electrode experiments of pure ionic liquids and ILs diluted with acetonitrile were also performed by impedance spectroscopy. The fibrous electrodes with the studied electrolytes showed different behaviors at analyzed frequency regions (see in Figure 8A–C).

The Nyquist plot of fibrous electrodes are presented in Figure 7A. For typical carbonous EDL electrodes, the −45° slope should be observed immediately after the semicircle, followed by a vertical line, which is almost parallel to the imaginary impedance axis (*Y*-axis) in the lower frequency region [44].

The closest results to EDLC behavior of the fibrous electrodes were achieved for quaternary ammonium-based electrolytes and with the EMIm-TFSI/ACN electrolyte, where a near-vertical line at a low frequency area was achieved. A clear semicircle was observed for EMIm-TFI/ACN in the high frequency region, which points to several possible reasons; for example, interfacial impedance occurring at the current collector/active material interface, mass transfer resistance inside microporous electrode at high frequencies, and charge transfer resistance inside macro/mesopores [44,48]. No clear semicircle was observed with other IL-based electrolyte solutions, probably due to the lower conductivity and higher viscosity of the electrolyte [33,49]. For pure IL-based electrolytes, high impedance was seen in the low frequency range, indicating that processes with limited mass transfer predominate at $f$ = 5 mHz, and even at 5 mHz; capacitive behavior was not ideal. This was caused by the much higher viscosity and much lower mobility of the electrolyte ions than in ACN-diluted solutions. A similar effect was also observed with EMImBF$_4$ by Zhang et al. when by diluting EMIm-BF$_4$ in ACN, the total resistance of the electrode decreased remarkably [43].

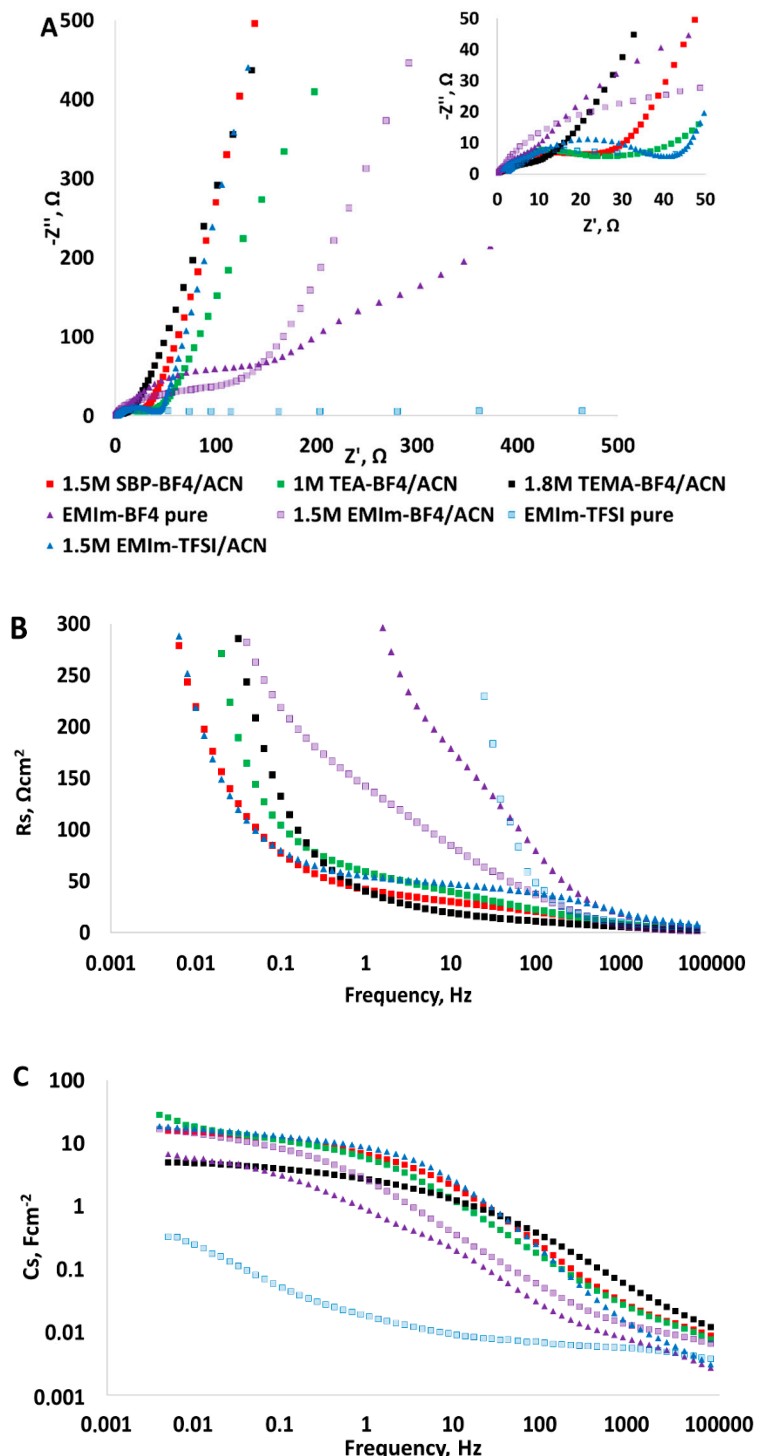

**Figure 8.** Electrochemical impedance spectroscopy (EIS) plots for electrolyte-carbon interface: (**A**) Nyquist plot, (**B**) $R_s$ as a function frequency and (**C**) $C_s$ as a function of frequency.

Series resistance and -capacitance, calculated by Equations (5) and (6) are shown in Figure 8B,C. According to the graph $R_s$ vs. $f$, the electrolytes can be divided into two groups: (1) quaternary salt solutions and (2) pure ionic liquids with EMIm-BF4/ACN. Interestingly, EMIm-TFSI/ACN behaved similarly to the first group of electrolytes. At high frequency region ($f > 1$ Hz) for the first group and ($f > 200$ Hz) for the second group electrolytes, $R_s$ values were much lower compared to the low-frequency region in Figure 7B. This was due to electrolyte ion migration speed and

their adsorption/desorption rate, which is maximum at low frequency of EDLCs [11]. The lowest resistance at the frequency region $10 < f < 100$ Hz was observed with TEMA-BF$_4$/ACN electrolyte. In addition, the resistance at 20 Hz increased in the following order: TEMA-BF$_4$ < SBP-BF$_4$ < TEA-BF$_4$ < EMIm-TFSI/ACN < EMImBF$_4$/ACN < EMIm-TFSI < EMIm-BF$_4$. This order correlates to the assumption that IL has a lower conductivity compared to organic electrolytes [12]. The lowest resistance among IL-based electrolytes of an electrode/electrolyte system was observed for EMIm-TFSI/ACN, as mentioned above. Also, the highest capacitance was achieved with the same IL-based electrolyte, EMIm-TFSI/ ACN. At high frequencies, the electrode material behaved more as a smooth surface and, therefore, electrolyte ions did not have enough time to migrate to carbon micropores, so the capacitance was very low. With decreased frequencies, the electrolyte ions started to migrate and adsorb into carbon pores, which resulted in an increase of the capacitance value. With a decrease in frequency ($f < 1$ Hz), capacitance reached the plateau for certain electrolytes. However, for pure IL-based electrolytes this plateau was not achieved even at very low frequencies ($f < 10$ mHz), due to their high viscosity, which can be observed in Figure 7C.

*3.4. Cycle-Life*

The cycle-life for electrospun fibrous CDC-based electrodes, was measured in a two-electrode set up by the CV method. The voltage scan rate was selected as 20 mV s$^{-1}$, due the electrode material previously showing steady capacitance behavior at this voltage scan rate. The cells were cycled between 0 V and 2.3 V. For the cycle-life measurements, SBP-BF$_4$/ACN and EMIm-TFSI/ACN electrolytes were selected. In the case of IL-based EMIm-TFSI/can, an exponential decrease in capacitance was observed during cycling, which may have been caused by the relatively high viscosity and high voltage scanning rate which disabled the effective ion transfer in the electrical double layer. Stable cycle stability was achieved with the SBP-BF$_4$/ACN electrolyte as shown in Figure 9. There was no significant capacitance loss (dC ~ 3%) and resistance increased by 21% during 3000 cycles.

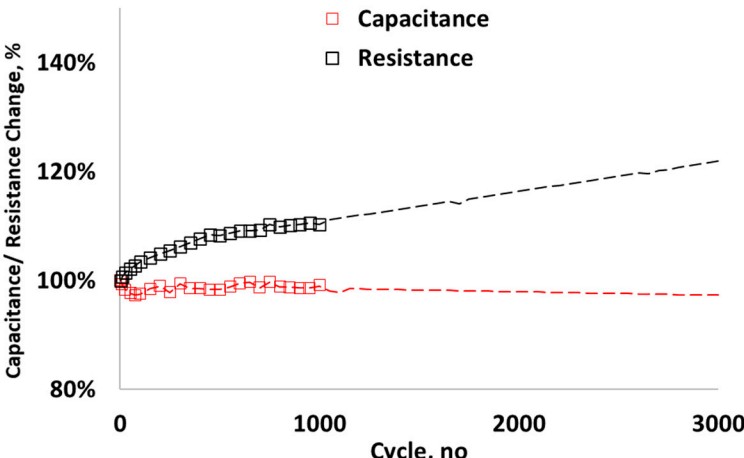

**Figure 9.** Cycle-life performance with SBP-BF$_4$/ACN electrolyte: capacitance retention and series resistance change during cycling.

## 4. Conclusions

The behavior of CDC containing flexible and mechanically durable fibrous electrodes was studied in various organic and IL-based electrolytes. The results of electrochemical evaluation showed that the widest potential range was achieved with a 1.5 M EMIm-TFSI/ACN electrolyte with a total voltage range of dU ≤ 3.5 V. This electrolyte also showed identical gravimetric capacitance 89.8 F/g for positively and negatively charged electrodes. Among quaternary ammonium salt-based electrolytes, the largest potential window was reached in 1.5 M SBP-BF$_4$/ACN–3.0 V, with a gravimetric

capacitance of 95.3 F g$^{-1}$ and 78.5 F g$^{-1}$ for positively and negatively charged electrodes, respectively. Electrochemical impedance spectroscopy showed that pure ionic-liquids had very high resistance and low DL-capacitance, making them practically unsuitable as electrolytes in electrospun materials. However, capacitance was significantly increased by diluting pure ionic liquid with acetonitrile; for 1.5 M EMIm-TFSI/ACN, the capacitance was increased 15-fold compared to pure ionic liquid. Stable cycle-life of electrospun CDC-based electrodes was achieved with organic SBP-BF4/ACN electrolyte with only a 3% capacitance loss.

**Author Contributions:** Data curation, S.M.; Investigation, S.M. and E.T.; Methodology, S.M.; Resources, V.V.; Supervision, M.A.; Validation, S.M. and I.K.; Writing—original draft, S.M.; Writing—review & editing, A.K. All authors have read and agreed to the published version of the manuscript.

**Funding:** This study was financially supported by the European Space Agency, ESA contract number 4000119258/16/NL/CBi "Fully electrospun durable electrode and the electrochemical double-layer capacitor for high frequency applications" and Estonian Ministry of Education and Research (IUT19-28).

**Acknowledgments:** We thank Sebastian Pohlmann and Ann Laheäär for fruitful discussions and consulting. Furthermore, we also thank senior researcher Valdek Mikli for SEM, Maike Käärik for BET and Can Rüstü Yörük for TGA analysis.

**Conflicts of Interest:** The authors declare no conflict of interest.

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
