# Peer review of "Electrochemical Evaluation of Directly Electrospun Carbide-Derived Carbon-Based Electrodes in Different Nonaqueous Electrolytes for Energy Storage Applications"

_carbon_

Round 1
Reviewer 1 Report
In this study, the authors are presenting their results about Electrochemical Evaluation of Directly Electrospun Carbide-derived Carbon-based Electrodes in Different Nonaqueous Electrolytes for Energy Storage Applications. Although their material seems to have good performance, the characterizations and their discussions are weak. There are missing characterizations, and most of the characterizations only report the results without a discussion. There is no cause and effect discussion and proper comparison with the literature. In this form, the paper cannot be published and need a major revision. Below, I have written some issues to be corrected. 1. Abstract should too small, describe more scientifically. The discussions are not satisfactory. For example, in the sentence, " Several studies suggest 14 investigating various electrolytes with asymmetric cells for casted electrodes, but not for the CDC based fibrous electrode materials. Benefits of the electrode with fibrous structure are relatively low 16 thickness (20 µm), flexibility and mechanical durability", what is the reason for that? Normally I would expect the opposite. Most of the paper is like this; it just gives the result and does not discuss why. 2. The clarity of the SEM shown in Fig.1 is not sufficient, in which add new scale bar as well as add different magnifications of all samples. I think the SEM should be considered for further modification. 3. The quality of the images is low. The texts in the images are so small that it very hard to read them. 4. The ‘The two semicircles appeared in the high-frequency region’ mentioned in the EIS analysis is inconsistent with the results shown in Fig.7a. It is recommended that the author should make a slight modification. Author should more described EIS and its relation to your work. 5. There are a few minor mistakes in the manuscript. The author should check them again carefully.. like all units , symbols, etc.. 6. Author should add XRD, and XPD for more confirmation.. 7. Any relation between capacitance and series resistance ??
Reviewer 2 Report
This manuscript presents a study of electrospun PAN-based carbon fibres doped with carbide-derived carbon (CDC). with respect to supercapacitor application using a variety of electrolytes. Since this journal is dedicated to carbon materials, I would prefer to see an even balance the parts dealing with carbon materials (rather short) and electrolyte aspects.
The manuscript could be improved further by a discussion of potential effects of the interaction of various electrolytes and the pore characteristics of the electrode (PSD, e.g.). Volumetric capacity of the electrode should be estimated for sake of comparison with practical capacitor electrodes.
